# Mechanisms of DNA Mobilization and Sequestration

**DOI:** 10.3390/genes13020352

**Published:** 2022-02-16

**Authors:** Kerry Bloom, Daniel Kolbin

**Affiliations:** Department of Biology, The University of North Carolina at Chapel Hill, Chapel Hill, NC 27599-3280, USA; kolbin@live.unc.edu

**Keywords:** genome mobility, gene clustering, transient cross-linking, polymer networks

## Abstract

The entire genome becomes mobilized following DNA damage. Understanding the mechanisms that act at the genome level requires that we embrace experimental and computational strategies to capture the behavior of the long-chain DNA polymer, which is the building block for the chromosome. Long-chain polymers exhibit constrained, sub-diffusive motion in the nucleus. Cross-linking proteins, including cohesin and condensin, have a disproportionate effect on genome organization in their ability to stabilize transient interactions. Cross-linking proteins can segregate the genome into sub-domains through polymer–polymer phase separation (PPPS) and can drive the formation of gene clusters through small changes in their binding kinetics. Principles from polymer physics provide a means to unravel the mysteries hidden in the chains of life.

## 1. Introduction

DNA breaks are beacons for recruiting DNA repair machinery and activating signaling pathways to put the brakes on cell cycle progression. DNA breaks are the Achilles’ heel of the genome, in the sense that they wreak havoc on the genome in the absence of repair. Genome instability in the form of mutagenesis and gross chromosomal rearrangements are just two of the consequences of inefficient or error-prone DNA damage repair. The diversity of repair pathways and pathway choice are dependent upon a myriad of factors including stage of the cell cycle, stage of development, ploidy, type of damage, spatial position in the nucleus, and spatial position in the chromosome, to name a few. In addition to the enzymology of repair and the type of DNA damage, there are significant compartmentalization constraints within the genome. Repair of DNA damage within heterochromatin [1,2], the nucleolus [3,4], and the centromere [5,6] are examples of compartments that contribute to repair pathway choice. Layered on these constraints is the motion of the DNA itself. The entire genome is mobilized following DNA damage [7,8,9,10,11]. While we lack a detailed understanding of the contribution to DNA repair afforded through enhanced genome mobility, principles from polymer physics help shed light on mechanisms responsible for sequestering and mobilizing sub-domains in the genome. 

Insight into genome compartmentalization and dynamics benefits from models describing long chain polymer motion such as DNA [12,13]. Stochastic simulations of entropy-driven, bead–spring polymer chain models account for many features of the dynamic properties of chromatin fibers, and have been responsible for significant advances in the past decade, elucidating higher-order genome organization. The challenge in the DNA damage and repair field is to marry the static DNA sequence-based perspective, the enzymology of DNA repair, and live cell measurements of DNA movement with the modeling of a dynamic bead–spring chromatin polymer. 

### 1.1. Modeling 

The dynamics of DNA are captured using bead–spring polymer models, where the chromosomes are represented by interacting beads connected via springs described by a worm-like chain (WLC) force law [14,15]. The springs can be thought of as chromatin “blobs” [16], rather than linear DNA segments (Figure 1).

Each blob is a region of the polymer over which the DNA behaves according to the thermal statistics. The size of the blob depends on the number of monomers, defined as N_k_ Kuhn lengths of length 2L_p_, where L_p_ is the persistence length of the chromatin fiber. Persistence length is a measure of a chain’s flexibility (L_p_ = length scale over which the chain is stiff, mathematically when the tangent vectors along the monomer of the chain become uncorrelated, L_p_ for DNA = 50 nm, ~150 bp). The dynamics of the segments inside each blob follow those of a random walk, forming what is called a Brownian bridge. The dynamics of the chain also follow a random walk that can be quantitated through mean square displacement. The size of the blobs represents the spatial extent over which the motion of the chromatin fiber is uncorrelated. Generalizing in this way allows the model to adapt to a range of useful scales. In one scale, the bead–spring chain represents the beads on a string-like chain of nucleosomes. At another scale, the bead spring chain represents genes, with each gene represented by a blob, strung on a chain with many other genes. Dynamics of the bead–spring chains are governed by a force balance. In the nucleus, forces controlling molecular interactions include van der Waals (attractive or repulsive), electrostatic (attractive or repulsive), steric (repulsive), hydrodynamic, entropic, and thermal forces. Attractive forces are captured through a spring-like force and repulsive forces by an excluded volume potential. Since the chromatin network is embedded in a highly viscous environment, hydrodynamic interactions are incorporated [13,17]. Principles from these Rouse models have been used extensively in the field to deepen our understanding of experimental observations of chromatin motion [9,10,13,18,19,20,21,22].

### 1.2. Spatial Segregation

In the span of the last several years, the field has identified two distinct modes of the compartmentalization of genomic sub-domains. One of these is through liquid–liquid phase separation (LLPS), the other through polymer–polymer phase separation (PPPS) [23]. Each of these mechanisms describe how non-contiguous regions of the genome can be aggregated in a reversible and tunable fashion. For LLPS, the condensates are driven by chemical properties of the DNA and/or RNA binding proteins (steric attraction of side chain, electrostatic interaction of charged groups). For PPPS, compartmentalization is driven by bridging or cross-linking multiple DNA loci through cohesin and condensin complexes. In the case of the rDNA, cross-linking between the rDNA gene repeats is sufficient to account for the segregation, morphology, and self-interaction of the nucleolus [17]. Cohesin, as well as transcriptional activators, can also drive PPPS as reported in the bridge-induced phase separation (BIPS) for budding yeast cohesin in vitro [24] and the switch–like transition from a relaxed to compact state of mouse heterochromatin [25].

### 1.3. Genome Mobility 

The genome is constantly moving, as are all components in the cell. What regulates the rate of genome movement? Firstly, let’s consider how we measure movement. On a long chain polymer, when we discuss the rate of movement, we typically refer to a DNA locus within the genome. The locus is marked by a fluorescent reporter system (FROS, fluorescent reporter–operator system). The locus behaves as predicted for a region along a long-chain polymer. This behavior was described by Rouse in 1953 [26], in which beads in a long chain polymer feel the random forces described above, but are confined due to their linkage to neighboring beads in the chain. Using mean square displacement (MSD) as the metric, the motion scales as t^1/2^ (Rouse mode) or t^2/3^ (Zimm mode) [27], in contrast to scaling with t^1^ for a freely diffusing bead. Thus, relative to a free bead, the polymer exhibits sub-diffusive motion. In addition, motion of a locus is confined due to constraints imposed by the nucleus (confined sub-diffusive motion) and the other chains (chromosomes). Second, we consider the forces that drive the motion. In addition to the molecular attractive and repulsive forces above, force on the bead is felt from the active motion of protein chaperones, remodelers, repair enzymes, RNA and DNA polymerases, and cytoskeletal motor proteins, to name a few [28,29,30,31,32]. As forces from protein collisions are random in direction, they can be observed by the increase in the plateau value of the MSD curve. Inside an active nucleoplasm, beads on a chain explore a greater area when compared to the situation where all active motion driven by ATP hydrolysis is removed (e.g., azide treatment) [32,33,34]. While the directed motion of chains has been reported [35], these are evolved mechanisms that are superimposed on motion, governed by the rules of polymer physics. 

The kinetics and density of cross-linking proteins provide another means of regulating the mobility of the genome, as well as localized sub-domains. In a non-intuitive way, cross-linkers have a profound impact on chain motion. As mentioned above, cross-linkers can promote PPPS. However, the kinetics of cross-linking (on and off rates) effect the clustering of genetic loci within the compartment, as well as the motion of the loci. Using the nucleolus as a model, we have shown that the timescale of binding kinetics has a measurable impact on the segregation and heterogeneity of gene clusters within the nucleolus [17,36]. In particular, fast kinetics of cross-linking (short t_½_) induces clustering of bead–bead contacts within the nucleolus, indicative of a structure-within-structure in nucleolus morphology, which homogenizes with slower binding (long t_½_). Fast kinetics induces the clustering of non-contiguous loci. The cross-linker is more mobile than the chain, such that when one cross-linker unbinds, another comes in before the DNA “gets away”. Likewise, the unbound cross-linker will quickly bind another loci, leading to clustering of loci over time (Figure 2A). 

Slow kinetics lead to a homogenization of loci (Figure 2B). Upon unbinding, the chain has time to diffuse (sub-diffusion, Rouse time, [27]) before contacting another cross-linker. This leads to the homogenizing of the loci, or declustering. From the perspective of the chain, in the case of the short-lived cross-linkers, chain motion is considerably constrained due to the clustering. In the case of long-lived cross-linkers, the chains are constantly moving and exhibit a greater degree of freedom and area explored relative to the short-lived cross-linking situation. Thus, the timescale of DNA motion is significantly influenced by the binding kinetics of cross-linkers and provides a source of genomic regulation. 

## 2. How Cohesin and Condensin Regulate Chromatin Motion

In deducing the function of the SMC (structural maintenance of chromosome) proteins cohesin and condensin in DNA repair, we need to understand their mechanism of action. The ring-like structure of the cohesin complex, the opening and closing of the rings, and proteolytic mechanisms have led to a paradigm of topological entrapment. In the case of cohesin and condensin, their ability to extrude DNA loops in vitro provides a mechanism to drive higher-order chromosome compaction and organization. In addition to entrapment and looping mechanisms however, alternative modalities such as hinge folding [37,38], in which the coiled coils bend and interact with the DNA helix (cohesin) [37], multimerization [39,40], or scaffolding [41] have been observed. These modalities provide flexibility and diversity in the functions for which SMCs contribute. 

There is considerable flexibility in the substrate (DNA) as well. A simple stepping motor in a molecular dynamics model will extrude loops as it traverses a floppy substrate [42]. As a “motor” steps, one head remains static and the other swivels to find the next closest binding site in the DNA chain. There is no rule that the closest site open to binding has to be adjacent on the linear polymer. The closest site is whatever region of the polymer has fluctuated to the vicinity of the free motor head, and can be displaced in linear space (i.e., with respect to the polymer chain). In this way, loops are extruded due to chain fluctuation and the binding and unbinding of the protein to the DNA. The ability of cohesin and condensin subunits to multimerize provides additional means to cross-link chains, yielding loops if the cross-links are intra-chromosomal, or a polymer network if linking different chromosomes [43]. 

Considering alternative SMC structure and substrate fluctuations may be instructive in situations such as DNA repair, where SMCs are recruited to sites of DNA damage in yeast [44,45] and humans [46,47,48] outside the unperturbed cycle of events. It has been proposed that cohesin recruitment at sites of double strand breaks promotes sister chromatid exchange for recombination-based repair pathways [49,50,51,52], as well as restraining movement [53]. The challenge is to reconcile cohesin’s role in recombination-based repair with the enhanced movement proposed to facilitate homology search [8,53,54,55]. When viewed from the perspective of a polymer network (Figure 1), increasing cohesin at sites of DNA damage without topological constraints such as ring binding will increase chain motion due to the homogenizing of the network. With fast binding cohesin (short t_½_), movement is restrained due to the formation of clusters. Increasing the local concentration of cohesin (effectively longer half-life in the model) will result in increased motion due to release of clusters/nodes. In contrast, studies depleting condensin have demonstrated an increase in motion [53,56]. These non-monotonic relationships are characteristic of polymer networks and their regulation (through L_p_, cross-linkers, etc.) [57]. Recruiting cohesin to the site of DNA damage releases DNA from a cluster/node, resulting in increased motion. Thinking of cohesin and condensin beyond the simple ring structures that entrap DNA, and at the level of chain dynamics and cross-linker status provides important insights into understanding the properties of these anastomosing systems of polymers, cross-linkers and slip-links thereby expanding their functions in chromosome dynamics and organization.

## 3. How Cohesin and Condensin Promote Polymer Phase Separation

While much attention has been focused on the role of disordered proteins in phase separation, the contribution of polymer–polymer phase separation (PPPS) through cross-linking is a robust means to sequester sufficient molecular species for metabolic processes [23,58]. These mechanisms have been explored from the theoretical side for decades [59] and describe how, in even a single chain, the formation of phase-separated regions can readily be generated. Not surprisingly, these PPPSs can do work associated with the displacement and reorganization of chromatin in several ways. A PPPS is a subdomain in an otherwise packed nucleus, surrounded by a depletion zone. By analogy to colloid particles in suspension, entropic depletion forces drive particles together due to the increase in volume in the system (and more freedom for the solute) when depletion zones of two particles overlap. These depletion forces provide a means to segregate subdomains that will find their thermodynamically favorable position, often being peripheral to the bulk of the chromatin. Forces from active processes, including transcription [58] and condensation [60], will also lead to chain displacement. As stated above, active transcription in the nucleolus is required to retain miscibility between the liquid and polymer phases [58]. Quail et al. [60] further demonstrated that condensed subdomains can expand by exerting force on the chain and reeling in additional polymer. It is evident that the ability to partition the genome into discrete domains that are soft or stiff, either through multivalent binding of DNA-associated proteins (dense but soft) or density and kinetics of cross-linking proteins (heterogeneous or homogeneous), leads to architectural changes that can be understood from the material properties of the components. 

We understand how cross-linkers can facilitate the partitioning of specific sub-domains (transcriptional units) from the mass of the genome, and that those sub-domains may or may not be contiguous. The next question is how these domains can be tuned for specific biochemistries. For this we turn to polymer physics, focusing on transitions between coils and globules (Figure 3). 

Polymers undergo a well-known phase transition, from a random coil to a globule, similar to the phase transition from a gas to a liquid. The size of the random coil is a function of the overall chain length (L_c_ = contour length) and the persistence length, which for DNA = 50 nm (150 bp). In the coiled state, monomeric units are solvated and repulse each other at short distances, leading to a swollen coil. The globule is a conformation in which the monomeric units are tightly packed (analogous to a liquid), decreasing the volume the polymer occupies. A difference between a polymer globule vs a liquid is that in the globule, monomers are linked. The nature of the solvent is sufficient to transition from a coil (good solvent) to a globule (bad solvent). The heterogeneous distribution of small molecules, including ions (e.g., Ca^++^) highlight the contribution of solvent as a regulator of polymer state [61,62,63], providing an additional modality in terms of polymer density for understanding how sub-domains can be built. As we explore the configuration of a long-chain polymer and the various means of regulation (cross-linking, globule–coil transition), heterogeneity along the chain, i.e., sub-domains, are the expected configuration, depending on the distribution and concentration of cross-linkers and solutes.

## 4. Physical Basis for the Spatial Segregation of Repair and Signaling Pathways

Condensin and cohesin are concentrated in two specialized regions of the genome, the nucleolus and the pericentromere [64]. There is ample evidence for the phase separation of both these regions [65,66], as well as for the sequestration of DNA repair machinery [4,5,67,68,69]. A third major compartment within the nucleus that is phase separated (via HP1 or transcriptional activators) and sequesters repair proteins within is the heterochromatin [1,2,25]. Recruitment of cohesin to sites of DNA DSBs provides a mechanism to create transient repair zones. Regulation of cohesin velocity, directionality, and on/off chromatin binding kinetics will impact the ability to sequester damaged zones and thus the efficiency of repair [70].

Homologous recombination is suppressed in the nucleoli and centromeres of both yeast and mammals [4,71,72]. In yeast, Rad52 exclusion from the nucleolus is dependent upon the SMC proteins, Smc5, 6 [4]. Likewise, Rad51 exclusion from the HP1 heterochromatin domain is dependent upon Smc5,6 [1]. The exclusion of HR proteins from compartments that contain repeat DNA sequences is understood as a means to prevent rearrangements between repeats and potential genome shuffling and instability. In yeast, damaged rDNA must migrate outside the nucleolus for Rad52 dependent repair [4]. It is likely that both the LLPS and PPPS properties of these sub-domains contribute to their specific biochemical properties by concentrating or excluding necessary repair factors. 

The centromere and pericentromere are also regions where there is spatial segregation of DNA repair pathways. In mammals, it has been shown that centromere is active in HR (homologous recombination) and NHEJ (non-homologous end-joining) throughout the cell cycle, while the pericentromere is active in NHEJ in G1, and NHEJ and HR in S/G2, indicative of differential regulation in these two proximal domains [6]. Double strand breaks travel to the periphery of the centromere domain, while breaks in the pericentromere remain and recruit Ku80 [6]. While the pericentromere of budding yeast lacks the repetitive elements characteristic of mammalian centromeres, DSBs in the pericentromere are preferentially repaired via NHEJ, while DSBs in the chromosome arm repaired via HR [5]. 

With respect to signaling, a non-canonical ATR pathway has been found to be enriched in mammalian centromeric chromatin, where it contributes to chromosome fidelity [68]. Other researchers have found that the ATR checkpoint is suppressed in the centromere in order to prevent checkpoint activation upon replication fork pausing [67]. Several key S-phase checkpoint proteins (Mrc1, Tof1, and Csm3) localize to replication forks, where they delay replication fork progression through the centromere [73] and recruit cohesin [74].

## 5. Summary

Polymer physics is a powerful guide for the behavior of a transient, responsive, cross-linked network such as the genome. One of the major paradigms is the disproportionate effect of weak interactions in these networks. Cross-linking kinetics can transform a homogeneous structure into one that is dominated by gene clusters (short-lived interactions favor clusters, [17,36]), decrease the self-healing time for a polymer network (shortest self-healing time when cross-links rapidly form and break [75]), and drive coil-to-globule transitions [76]. 

Polymer networks with reversible bonds are self-healing (i.e., enzyme-assisted healing property of the genome). When a polymer network is damaged, different healing events are observed, depending on the timescale (Figure 4). Self-healing, where a broken chain rapidly seeks a repair partner, is an immediate response to damage. However, if there is long wait time before repair can occur, the broken chain equilibrates (i.e., gets homogenized) and a different repair pathway, known as self-adhesion, predominates. The point is that there is every reason to expect mechanisms that influence chain motion, such as those described above, to have profound consequences on DNA repair pathway choice. From a polymer perspective, self-healing may be the physical analog to non-homologous end-joining, while self-adhesion is analogous to HR. 

In addition to the role of cross-linkers, slip links, or molecular pulleys, can contribute to compartmentalization of sub-domains. Slip-links provide a means to maintain shape over a range of solute conditions [78] and lead to non-linear rheological properties in a polymer network [79]. The agreements between bead–spring models and experimental data highlight the necessity to consider transient roles of proteins such as SMC’s (cohesin, condensin, SMC 5,6) that can tune polymer networks via cross-linking or as slip links in a rapid response to DNA damage and additional environmental insults.

The polymer can also be modified in ways that lead to changes in mobility [80]. Mine-Hattab et al., found that the anomalous MSD diffusion exponent (alpha) is smaller at short time scales, consistent with models predicting stiffening of the chain [9,80]. Epigenetic modification can also influence chain persistence length [81], providing yet another means of regulation. The most extreme case of histone regulation is the degradation of histone protein following DNA damage, as a driver of increased chromatin motility [82,83]. To better understand the consequences of histone depletion from the perspective of the DNA chain, we have introduced histones and their dynamic turnover into the fluctuating bead–spring chains [57,84]. Using in vivo and in silico approaches we found that both histones and condensin cross-linkers were required in silico in order to quantitatively match the dynamic chromatin compaction observed in vivo [57]. An exact experimental correlate of our bead–spring simulation was obtained through analysis of a model-matching, fluorescently labeled, circular chromosome in live yeast cells. Experimentally observed chromosome compaction and variance in compaction were reproduced only with tandem interactions between histone and condensin, not from either individually. Furthermore, several non-intuitive properties of the polymer have emerged from implementation of histone and condensin function. There is a non-monotonic relationship between persistence length and mean square displacement in the presence of histone and condensin. This has important consequences in terms of inferring mechanisms based on simpler realizations of these bead–spring models. Second is the mechanical coupling between condensin function and tension on the bead–spring chain. High regions of tension on the chain restrict the loop-extrusion activity of condensin, forcing condensin to idle in place [57]. Condensin drives fluctuations by actively extruding loops along the chromosome in segments of low tension. Histones, on the other hand, compact the large loops formed by condensin, while generating the tension that modulates loop generation. These two chromatin-binding proteins lead to a compact, highly fluctuating chromosomal structure. A comprehensive review of histone modifications and consequences with respect to DNA damage can be found in Ferrand et al. [85].

A unifying perspective that emerges from the polymer physics is to recognize that changes in mobilization may reflect underlying changes in viscoelasticity, including small molecule kinetic processes that drive and sustain the polymeric system out of equilibrium. Proteins such as fibrin are extensible and elastic, with rapid coil (worm-like chain extensibility) and recoil. High-speed analysis of fibrin recoil events reveal extension and entropic recoil of unstructured regions to be the source of recoil, rather than reversible protein unfolding [86]. The genome, like fibrin, may be optimized mechanically for elastic energy storage, enabling repeated cycles of stretching and recoil in response to pico-Newton scale mechanical perturbations. This feature endows the genome with an exquisite sensitivity to force and a means to endure multiple cycles of pulling and prodding, consisting of transitions between states of organization and associated viscoelastic properties.

## Figures and Tables

**Figure 1 genes-13-00352-f001:**
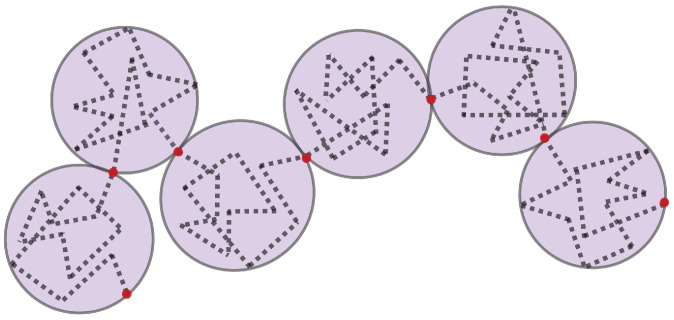
Bead spring representation of a chromosome. The diameter of each ‘blob’ is the length of a spring. The dotted line represents DNA. DNA within each blob follows a random walk. The Brownian bridge is between the red beads.

**Figure 2 genes-13-00352-f002:**
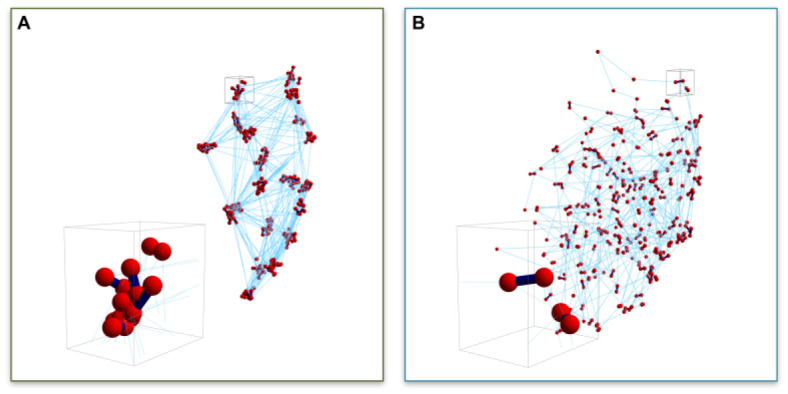
Snapshots of 3D nucleolus simulations. Bead distributions for (**A**) ton = 0.09 s and (**B**) ton = 90 s. Red symbols (spheres) represent bead positions, dark blue segments (lines) represent transient crosslinks between beads both inter- and intra-chain, light blue (thin) lines represent intra-chain neighboring bead connections. Inserts in (**A**,**B**) are blow-ups of small volumes around bead clusters. (Figure 7 from [17]). (Reprinted with permission from Ref. [17]. Copyright 2017 Nucleic Acids Research, Oxford University Press).

**Figure 3 genes-13-00352-f003:**
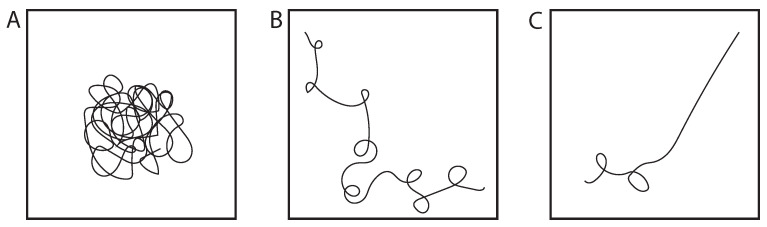
Globule-coil-rod transition. (**A)** The globule is a condensed state of the polymer, reflecting poor solvent conditions. (**B**) In a good solvent, the chain is a swollen state, known as a polymer coil. (**C**) In living systems, these coils can exhibit regions of stiffening (rod state) for biological functions [9,57].

**Figure 4 genes-13-00352-f004:**
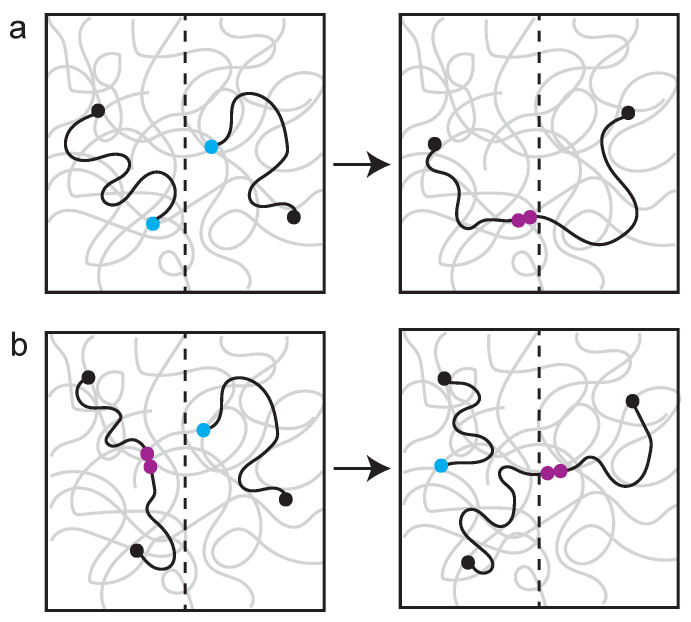
Examples of self-healing versus self-adhesion in a polymer network. The network is visualized as squiggly gray lines, broken chains are visualized as black squiggly lines. A broken end is indicated with a light-blue ball. (**a**) Self-healing: fast kinetic where two breaks (blue) rapidly find one another and heal (maroon). (**b**) Self-adhesion: slow kinetics where a break (blue) has to find a partner and exchange. The system has time to re-equilibrate and the broken chain exchanges with an ectopic site (making a new healed site (maroon)). Adapted from Rubinstein [77].

## Data Availability

Not Applicable.

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
