# Peer review of "Mechanisms of DNA Mobilization and Sequestration"

_genes, 2022, doi:10.3390/genes13020352_

Round 1

Reviewer 1 Report

This review entitled « Mechanisms of DNA Mobilization and Sequestration upon DNA Damage », by Kerry Bloom relies on principles from polymer physics to decipher the importance of the phase separation driven by cross-linking proteins on genome organization. This review is well written and tackles a very important question with a strong focus on the role of PPPS on DNA interactions. On the other hand, the DNA damage part of the review is not developed enough since it does not take into consideration the temporality of repair, the differences observed between repair of unique and multiple DNA breaks or between repairable and unrepairable DSBs. The underdevelopment of this part of the review makes the title somewhat misleading.

This review begins with a concise yet pedagogic explanation of the basis of polymer-polymer phase separation (PPPS). However, since this review focuses mainly on the role of cohesins on PPPS, it is surprising that it does not reference recent experimental work reporting bridging induced phase separation by cohesins.
(Ryu, J.-K.; Bouchoux, C.; Liu, H.W.; Kim, E.; Minamino, M.; de Groot, R.; Katan, A.J.; Bonato, A.; Marenduzzo, D.; Michieletto, D.; et al. Bridging-induced phase separation induced by cohesin SMC protein complexes. Sci. Adv. 2021, 7, eabe5905.).

While describing the Rouse model, the review should cite pioneering work done in yeast for different chromosomal loci.

This review also tackles the role of cohesins in spatial segregation during DSB repair. However, it would be interesting to encompass recent data proposing that cohesin enrichment at DSB drive a one-side loop extrusion and create a specific compartment near the break (Arnould, C.; Rocher, V.; Finoux, A.-L.; Clouaire, T.; Li, K.; Zhou, F.; Caron, P.; Mangeot, P.E.; Ricci, E.P.; Mourad, R.; et al. Loop Extrusion as a Mechanism for Formation of DNA Damage Repair Foci. Nature 2021, 590, 660–665.)

The role of HP1 in mouse chromatin compartimentalization has recently been assessed and should be taken into account since it illustrates that polymer collapse can contribute to the formation of chromatin compartment (Erdel F, Rademacher A, Vlijm R, Tunnermann J, Frank L, Weinmann R, Schweigert E, Yserentant K, Hummert J, Bauer C, et al. Mouse heterochromatin adopts digital compaction states without showing hallmarks of HP1-driven liquid-liquid phase separation. Mol Cell. 2020;78(2):236–249. doi: 10.1016/j.molcel.2020.02.005).

Although modifications affecting the polymer mobility such as stiffening of the chain and epigenetic modifications are evoked at the end of the manuscript, this review could benefit from a more detailed perspective of their role in the mechanisms of DNA mobilization. Notably, the contribution of histone modifications to long-range interactions should be developed. 

Author Response

This review entitled « Mechanisms of DNA Mobilization and Sequestration upon DNA Damage », by Kerry Bloom relies on principles from polymer physics to decipher the importance of the phase separation driven by cross-linking proteins on genome organization. This review is well written and tackles a very important question with a strong focus on the role of PPPS on DNA interactions. On the other hand, the DNA damage part of the review is not developed enough since it does not take into consideration the temporality of repair, the differences observed between repair of unique and multiple DNA breaks or between repairable and unrepairable DSBs. The underdevelopment of this part of the review makes the title somewhat misleading.

We thank the review for this comment. We agree with the assessment and have changed the title to read:

Mechanisms of DNA Mobilization and Sequestration.

This highlights the polymer perspective of the review and diminishes expectations related to DNA damage, as pointed out by reviewer #2 as well.

This review begins with a concise yet pedagogic explanation of the basis of polymer-polymer phase separation (PPPS). However, since this review focuses mainly on the role of cohesins on PPPS, it is surprising that it does not reference recent experimental work reporting bridging induced phase separation by cohesins.
(Ryu, J.-K.; Bouchoux, C.; Liu, H.W.; Kim, E.; Minamino, M.; de Groot, R.; Katan, A.J.; Bonato, A.; Marenduzzo, D.; Michieletto, D.; et al. Bridging-induced phase separation induced by cohesin SMC protein complexes. Sci. Adv. 2021, 7, eabe5905.).

Added to Spatial Segregation:

Cohesin as well as transcriptional activators can drive PPPS as reported in the bridge-induced phase separation (BIPS) for budding yeast cohesin in vitro [24] and the switch–like transition from a relaxed to compact state of mouse heterochromatin [25].

While describing the Rouse model, the review should cite pioneering work done in yeast for different chromosomal loci.

Citations incorporated below: Principles from these Rouse models have been used extensively in the field to deepen our understanding of experimental observations of chromatin motion  [9,10,13,18-22].

This review also tackles the role of cohesins in spatial segregation during DSB repair. However, it would be interesting to encompass recent data proposing that cohesin enrichment at DSB drive a one-side loop extrusion and create a specific compartment near the break (Arnould, C.; Rocher, V.; Finoux, A.-L.; Clouaire, T.; Li, K.; Zhou, F.; Caron, P.; Mangeot, P.E.; Ricci, E.P.; Mourad, R.; et al. Loop Extrusion as a Mechanism for Formation of DNA Damage Repair Foci. Nature 2021, 590, 660–665.)

We added the following (4, Physical basis…)

Recruitment of cohesin to sites of DNA DSBs provides a mechanism to create transient repair zones. Regulation of cohesin velocity, directionality and on/off chromatin binding kinetics will impact the ability to sequester damaged zones and thus the efficiency of repair [70].

The role of HP1 in mouse chromatin compartimentalization has recently been assessed and should be taken into account since it illustrates that polymer collapse can contribute to the formation of chromatin compartment (Erdel F, Rademacher A, Vlijm R, Tunnermann J, Frank L, Weinmann R, Schweigert E, Yserentant K, Hummert J, Bauer C, et al. Mouse heterochromatin adopts digital compaction states without showing hallmarks of HP1-driven liquid-liquid phase separation. Mol Cell. 2020;78(2):236–249. doi: 10.1016/j.molcel.2020.02.005).

Section following Spatial segregation

Cohesin as well as transcriptional activators can also drive PPPS as reported in the bridge-induced phase separation (BIPS) for budding yeast cohesin in vitro [24] and the switch–like transition from a relaxed to compact state of mouse heterochromatin [25].

We added an additional clause in section 4, 1st paragraph.

A third major compartment within the nucleus that is phase separated (via HP1 or transcriptional activators) and sequesters repair proteins is the heterochromatin [1,2,25]

Although modifications affecting the polymer mobility such as stiffening of the chain and epigenetic modifications are evoked at the end of the manuscript, this review could benefit from a more detailed perspective of their role in the mechanisms of DNA mobilization. Notably, the contribution of histone modifications to long-range interactions should be developed. 

We thank the reviewer for pointing out this omission and add the following section

The most extreme case of histone regulation is the degradation of histone protein following DNA damage as a driver of increased chromatin motility [82,83]. To better understand the consequences of histone depletion from the perspective of the DNA chain, we have introduced histones and their dynamic turnover into the fluctuating bead–spring chains [57,84]. Using in vivo and in silico approaches we found that both histones and condensin crosss-linkers were required in silico in order to quantitatively match dynamic chromatin compaction observed in vivo [57]. An exact experimental correlate of our bead-spring simulation was obtained through analysis of a model-matching fluorescently labeled circular chromosome in live yeast cells. Experimentally observed chromosome compaction and variance in compaction were reproduced only with tandem interactions between histone and condensin, not from either individually. Furthermore, several non-intuitive properties of the polymer emerged from implementation of histone and condensin function. There is a non-monotonic relationship between persistence length and mean-square displacement in the presence of histone and condensin. This has important consequences in terms of inferring mechanism based on simpler realizations of these bead-spring models. Second is the mechanical coupling between condensin function and tension on the bead-spring chain. High regions of tension on the chain restrict the loop-extrusion activity of condensin, forcing condensin to idle in place [57]. Condensin drives fluctuations by actively extruding loops along the chromosome in segments of low tension. Histones, on the other hand, compact the large loops formed by condensin while generating the tension that modulates loop generation. These two chromatin-binding proteins lead to a compact highly fluctuating chromosomal structure. A comprehensive review of histone modifications and consequences with respect to DNA damage can be found by Ferrand et al., [85].

Reviewer 2 Report

The current review work is one of the interesting topics authors have tried to review. As a reviewer I would like some in vitro or in vivo relevance of work. Please look for my comment.

Authors' correlation of addressing DNA damage dynamics bead spring models also need to be addressed with context of bench work finding and tools available for base exchange repair (BER or NER) or double strand break repair (DSB). Authors can find various examples of XPA, XPB, XPG or XPC mutant’s reate repairs (many Cisplatin based DNA adducts’ s) literature knowledge for NER and or mutational study work on DSB (look for Maria Jasin work of ISec based quantitative dynamics) of RAD51, ATM, ATR, BRACA etc deleted and rate of DSB repair. If possible, please make figures (not required if you don’t have enough material). Hope this incorporation will help the paper as a landmark.

Author Response

The current review work is one of the interesting topics authors have tried to review. As a reviewer I would like some in vitro or in vivo relevance of work. Please look for my comment.

To address the major concern of this reviewer and reviewer #1, we changed the title to “Mechanisms of DNA mobilization and sequestration”, that better reflects the focus on the behavior of the DNA chain, rather than biochemical details of DNA repair and repair machinery.

Authors' correlation of addressing DNA damage dynamics bead spring models also need to be addressed with context of bench work finding and tools available for base exchange repair (BER or NER) or double strand break repair (DSB). Authors can find various examples of XPA, XPB, XPG or XPC mutant’s reate repairs (many Cisplatin based DNA adducts’ s) literature knowledge for NER and or mutational study work on DSB (look for Maria Jasin work of ISec based quantitative dynamics) of RAD51, ATM, ATR, BRACA etc deleted and rate of DSB repair. If possible, please make figures (not required if you don’t have enough material). Hope this incorporation will help the paper as a landmark.

We appreciate the reviewer’s comments, but the detailed biochemistry of specific DNA repair is beyond the scope of this review. Our intent was to highlight the behavior of the DNA chain and for the field to incorporate the non-intuitive aspects of polymer behavior into detailed biochemical models.  
